# Resistive-Based Micro-Kelvin Temperature Resolution for Ultra-Stable Space Experiments

**DOI:** 10.3390/s23010145

**Published:** 2022-12-23

**Authors:** David Roma-Dollase, Vivek Gualani, Martin Gohlke, Klaus Abich, Jordan Morales, Alba Gonzalvez, Victor Martín, Juan Ramos-Castro, Josep Sanjuan, Miquel Nofrarias

**Affiliations:** 1Institut de Ciències de l’Espai (ICE,CSIC), Campus Universitat d’Autonoma de Barcelona, Carrer de Can Magrans s/n, 08193 Cerdanyola del Vallès, Spain; 2Institut d’Estudis Espacials de Catalunya (IEEC), Gran Capità, 2-4, Ed. Nexus, 08034 Barcelona, Spain; 3German Aerospace Center (DLR), Robert-Hooke-Str. 7, 28359 Bremen, Germany; 4Departament d’Enginyeria Electrònica, Universitat Politècnica de Catalunya, 08034 Barcelona, Spain

**Keywords:** temperature sensing, resistive sensors, space technologies, low frequencies, gravitational wave detection

## Abstract

High precision temperature measurements are a transversal need in a wide area of physical experiments. Space-borne gravitational wave detectors are a particularly challenging case, requiring both high precision and high stability in temperature measurement. In this contribution, we present a design able to reach 1 $\mu K/\sqrt{\mathrm{Hz}}$ in most of the measuring band down to 1 mHz, and reaching 20 $\mu K/\sqrt{\mathrm{Hz}}$ at 0.1 mHz. The scheme is based on resistive sensors in a Wheatstone bridge configuration which is AC modulated to minimize the 1/f noise. As a part of our study, we include the design of a test bench able to guarantee the high stability environment required for measurements. We show experimental results characterising both the test bench and the read-out, and discuss potential noise sources that may limit our measurement.

## 1. Introduction

High precision temperature measurements are becoming a transversal need in a wide variety of fields spanning both applied and fundamental physics. In most cases, high precision goes together with the demand for highly stable environments that can guarantee long-term operation of the experiment, as, for instance, is the case for exoplanet detection instruments [1]. Moving these experiments to space is a natural choice when trying to reach higher stability. Indeed, an increasing number of missions are seeking the stability provided by in-orbit experiments as a key requirement to achieve their scientific objectives [2,3,4,5]. Among space missions, gravitational wave detectors such as LISA [6] represent a particularly challenging area for temperature sensing, the main reason being that these observatories are designed to achieve their top sensitivity in the millihertz frequency regime. In these ultra-stable operations regimes, temperature fluctuations can perturb scientific measurements through a wide variety of phenomena, including thermal induced forces applied directly to the test mass and temperature-induced path length variations in the interferometers [7,8,9,10].

In recent years, there has been an increasing interest in the development of new technologies able to attain high temperature resolutions. Optical metrology experiments have demonstrated temperature precisions of 80 $\mathrm{nK}/\sqrt{\mathrm{Hz}}$ at 100 Hz by means of whispering gallery mode resonators [11], and even 3.8 $\mathrm{nK}/\sqrt{\mathrm{Hz}}$ at 1 Hz using more complex optical setups [12]. In parallel, nano-fabrication methods have been used to design temperature sensors based on micro-cantilevers or membranes [13], achieving a precision in the order of 10 $\mu K/\sqrt{\mathrm{Hz}}$ above 1 Hz. Despite these new developments, resistive-based systems are the preferred devices in space missions thanks to their reliability and long heritage. Among these, Johnson noise thermometers require cryogenic operation temperatures in
order to reach micro-Kelvin resolutions [14], while AC-modulated Wheatstone bridge schemes have already shown performance of 10 $\mu K/\sqrt{\mathrm{Hz}}$ at room temperature, a competitive
figure of merit that compares with proposed nano-fabricated techniques [15,16]. Morevoer, the former have extended the measuring bandwidth down to very low frequency (ultra-stable) regimes. This is the case for the temperature diagnostics subsystem used for LISA Pathfinder (LPF), which has shown an in-orbit perfomance of 10 $\mu K/\sqrt{\mathrm{Hz}}$ down to 1 mHz, an extremely challenging ultra-stable regime [17,18]. To the best of our knowledge, this represents the highest precision at very low frequencies ever measured either on the ground or in space.

In this article, we present an enhanced version of the temperature diagnostics subsystem used in LISA Pathfinder. The proposed design builds upon a previous knowledge and aims to improve the temperature measurement performance by one order of magnitude, reaching 1 $\mu K/\sqrt{\mathrm{Hz}}$ down to 1 mHz.

## 2. Setup Description

Our measurement subsystem has to face extremely demanding requirements in terms of high precision, range, and stability. It is worth stressing that the definition of an environment that ensures micro-Kelvin level temperature fluctuations at frequencies below millihertz is a challenging activity in itself. For this reason, when considering the design of our measurement subsystem, we considered the test bench as an integral part. In the following, we describe the design of both the test bench and the temperature measurement subsystem.

### 2.1. Low-Frequency Test Bench Design

Temperature fluctuations are hard to isolate in the very low frequency regime. While the heat transfer equations allow for easy suppression of relatively fast temperature changes by adding layers of low thermal conductivity material, the same scheme does not apply when trying to suppress longer duration perturbations. In the electrical analogy, when designing a thermal isolator one is forced to work with thermal low pass filters. Moreover, the lower the cut-off frequency of the filter, the higher the required equivalent mass of the insulator. To overcome this limitation, experiments requiring runs that can span from days to months, which can be necessary in cases ranging from gravitational detection [19] to fundamental physics [5,20] and frequency references [12,21], researchers usually resort to methods implicating multiple shields, which operate as a cascade of thermal low pass filters.

Analogously, our test bench design relies on a vacuum chamber housing a set of passive thermal shields. A distinctive characteristics of our design is that it includes a set of thermolectric cooling Peltier elements. These units allow the working points of the test bench between +10 °C and +40 °C to be changed while at the same time being used as an active control. The Peltier elements use heat pipes connected to the vacuum chamber lid to dissipate the heat generated when cooling below room temperature. In order to increase the cooling efficiency, the vacuum chamber lid is in turn cooled by means of a water-cooled system.

The design of the whole system is shown in Figure 1. Inside the chamber, three aluminium 2 mm-layers host the inner aluminium block where the temperature sensors are located. The Peltier elements are attached to the outermost layer of the aluminium shields. PEEK spacers (three between the layers) are used only in the bottom surfaces of the shields to minimise the conductive coupling.

Special care was taken in the design of the thermal shields. The design driver of these elements is the need to guarantee a stability of $100\;\mathrm{nK}/\sqrt{\mathrm{Hz}}$ down to 1 mHz. This requirement is set in order to guarantee that the temperature stability in the aluminium core at center of the shields is one order of magnitude better with respect to the performance of our measurement system. Following [22], the system is modelled taking into account both radiative and conductive links. According to this model, with an attenuation of almost six orders of magnitude at 1 mHz, the design can provide the required $100\;\mathrm{nK}/\sqrt{\mathrm{Hz}}$ at 1 mHz for a worst-case scenario in which the external lab temperature fluctuates at $100\;\mathrm{mK}/\sqrt{\mathrm{Hz}}$ at 1 mHz.

### 2.2. Sensor Readout Electronics Design

The main drivers of this project regarding the sensor read-out are twofold: first, to achieve a realistic scaled-down and COTS (Commercial Off-The-Shelf) component-based prototype for missions such as LISA, and second, to present a technology demonstrator able to reach the very demanding 1 μK/$\sqrt{\mathrm{Hz}}$ noise floor for frequencies from 0.1 Hz down to 1 mHz, from where it relaxes by a factor of 1/*f* until 0.1 mHz.

At these very low frequencies, the noise spectrum is dominated by pink noise (i.e., 1/*f* noise), which is typically much higher than the white noise limit found at higher frequencies. Hence, it becomes crucial to move the response of our system out of this lower-frequency region to a comparatively higher-frequency region where either the pink noise is low enough or we are already limited by the uniform white noise. This is the basic working principle of the lock-in amplifier or chopping technology used in multiple applications [23]. Therefore, as we discuss later in this section, we need to generate a modulation signal in order to move the response of the sensor to a higher frequency.

The first key decision is the selection of the sensors, which in our case are resistive sensors. To the best of our knowledge, this is the only technology currently available that fit the resource requirements and allows the number of sensors to be scaled up in a reasonable way. Specifically, the resistive sensors selected were NTC (Negative Temperature Coefficient) thermistors, which provide the highest sensitivity of all resistive sensors while being small in terms of both size and mass. Furthermore, NTCs are radiation immune for the expected levels inside a spacecraft, very robust regarding launch acceleration and spacecraft vibration levels, and can work in most temperature ranges, including cryogenics. Finally, they are a technology with a long flight heritage and proven reliability. Their downside is their magnetic material content; however, this effect can be made negligible by degaussing, as demonstrated for LPF [24] and their low but not negligible power dissipation below the hundreds of micro-watts.

Among the different dispositions used for reading the sensor value, we selected the Wheatstone bridge configuration (Figure 2), as it allows for reduction of the noise contribution by maximizing the gain of the chain at the first active stage [17]. In this configuration, the fixed resistance on the sensing arm resistor divider ($R_{c}$ in Figure 2) is set as close as possible to the nominal value of the thermistor at the center of the measurement scale in order to maximize the sensitivity.

In a Wheatstone bridge configuration with a voltage amplitude modulation, as in our case, the sensitivity of the system is equal to [17]:(1)$$s_{wb}=\frac{dV_{o}}{dT}=\frac{\beta V_{i}}{T^{2}}\frac{RR_{\mathrm{NTC}}\left(T\right)}{{\left(R+R_{\mathrm{NTC}}\left(T\right)\right)}^{2}}$$
where $V_{i}$ is the RMS (Root Mean Square) value of the voltage modulation applied on the Wheatstone bridge, *T* is the temperature in Kelvin, β is the NTC thermistor characteristic temperature curve, $R_{\mathrm{NTC}}$ is the temperature-dependent resistance of the thermistor, and *R* is the fixed resistor of the sensing arm. As can be seen, its behaviour is clearly nonlinear over the temperature measurement range. It is important to notice that in this document we mostly use the exponential equation of the NTCs for the theoretical expressions and refer to its β parameter, while in the actual implementation we use the Steinhart–Hart A, B, and C parameters. The reason for this is that while our absolute accuracy requirements are not very demanding (±0.1 K), the Steinhart–Hart equation provides better accuracy, which is required for the full range we are covering. Nonetheless, for the theoretical expressions the exponential approximation is good enough and more convenient for compact notation.

From Equation 1, that is, the sensitivity expression of the Wheatstone bridge, we can extract the following design drivers:The sensitivity increases with the voltage applied on the sensor; the obvious downside is that the power dissipation on the sensor itself does as well, leading to perturbations at the sensed location.The sensitivity does not change with the thermistor resistance.A higher β parameter increases the sensitivity without any downside.

Due to this last point, we decided to use 30 kΩ NTC thermistors with a β coefficient around 4200 K, as these specifications present one of the largest β values along with a resistance that is the lowest among the thermistors with this β. Keeping this resistance low, as shown in the later sections, allows us to minimize the current noise related to the later active input stage. Furthermore, it reduces the effect of inductive coupling due to external interference, which may be become relevant for long wires. Finally, it is important to note that thermistors have large manufacturing yields compared with, for instance, Platinum RTDs (Resistance Temperature Detectors). Hence, each sensor needs to be characterized and calibrated individually.

### 2.3. Read-Out Implementation

The actual implementation read-out consists of the elements in Figure 3, which we describe in detail below.

The first element is the DAC8831 Digital-to-Analog Converter (DAC) (Figure 2), which is controlled by an MCU (microcontroller unit) that runs the read-out software and generates the modulation stimulus signal. The voltage reference of the DAC and ADC (Analog-to-Digital Converter) are the same, that is, the system is ratiometric. Following this, the ADA4945-1 Fully Differential Amplifier (FDA) transforms the single-ended signal to a differential signal without any DC component. This balanced driving ensures that the common-mode and DC applied voltage is minimized, which is necessary to avoid an increase in the heat generated at the sensor. The FDA acts as an active low-pass filter, removing the stairs-like signal related to updating the output of the DAC. Because this is done each time a new sample is received from the ADC, this filter is a low-pass one with a kilohertz cut-off frequency.

The output of the Wheatstone bridge undergoes a differential amplification with a fixed gain. This amplification step is implemented by means of two operational amplifiers (one OPA2189) following a noise reduction criteria. Because the sensor impedance is relatively large, at tenths of a kilo-ohm equivalent input resistance, the key design driver for this amplification step is the current noise density. In order to keep the overall noise as low as possible in the 10 °C to 40 °C range, in this stage the gain is set to the maximum value. The output of the difference amplification remains unbalanced, however, and there is low common-mode due to the low gain. By adding an ADA4645-1 fully differential amplifier with a gain of one, we can recover the balanced driving of the ADC while adding the required common-mode for the selected ADC.

In the final step of the analogue chain, the ADC converts the analog value into a digital one and sends it to the MCU. The component selection for the ADC is critical, as the options for a space-qualified multi-channel high-resolution ADC are scarce. Because one of the main drivers of our approach is a scalable design able to sense at least ten sensors, we selected the ADS1278, which is an eight-channel simultaneous sampling ADC with 24-bit resolution. Furthermore, it has a space-qualified equivalent part, the ADS1278-SP.

#### 2.3.1. Measurement Principle

As introduced before, we need to generate a modulation signal in order to move the sensor response to a higher frequency region with a lower noise floor. The selection of the frequency is a trade-off between the 1/*f* noise, the desired measurement bandwidth, and the effects introduced by the wires to the sensors. This last point can become highly relevant in missions such as LISA, where it the wires can be expected to be from a few to tens of meters. The effect of the parasitic capacitances of wires, boards, and connectors, along with their high dependence on the ambient temperature, can severely impact the measurements, and the seriousness of this issue increases with the modulation frequency. These effects can be minimised in part by analysing only the in-phase component, although this does not completely avoid the problem. Therefore, we characterized the noise of the full chain using the zero calibration position of the multiplexer, which has a high-stability resistor with a similar value to that of the sensor at the center of the scale. As can be seen in Figure 4, the pink noise corner frequency is about 2 Hz. This is because the operational amplifiers and ADC used here are chopper-stabilized; therefore, in principle their pink noise contribution is negligible. The only element contributing is the FDA; as we describe later, this noise is divided by the gain of the input stage, and appears only at very low frequencies. Considering all these issues, we decided to use a nominal modulation frequency of about 10 Hz for our measurements.

For the results presented in this work, we used a square signal modulation. The square signal has a number of advantages with regard to a sinusoidal signal. First, it is immune to the phase noise of the modulation. Second, the computation required by the demodulation is much simpler. Even if it is not used in this application, its usage is compatible with multiplexing sensors. Finally, it is useful from a practical implementation point-of-view, as the sensitivity increases with the RMS voltage of the applied signal, as seen in Equation (Equation 1). The RMS voltage of the square wave is equal to its amplitude, in contrast to a sinusoidal wave, where it is equal to the amplitude divided by the square root of two. Hence, with the same modulation signal amplitude we can obtain a better sensitivity with a square wave than with sinusoidal modulation. However, a square wave has downsides as well. For instance, when demodulating the square wave, both the fundamental frequency noise and the noise of the harmonics are folded back into the measurement. Furthermore, due to the presence of long wires to the sensors, a spurious signal can be presented by the harmonics of the square wave. Hence, for the square wave we need to find a frequency for which both the fundamental frequency and its first harmonics are free of such spurious signals. In contrast, for a sinusoidal signal we only need to have a single frequency with a bandwidth as narrow as our measurement band of interest that is free of spurious signals.

For simplicity, for the noise analysis we use a linear approximation of the voltage-to-temperature dependence:(2)$$T_{m}=\frac{V_{\mathrm{ADC}}}{V_{\mathrm{wb}}\;G\;s_{\mathrm{wb}}}+T_{0}$$
where $T_{m}$ is the temperature measured by our system, $V_{\mathrm{ADC}}$ is the voltage measured by the ADC, $V_{\mathrm{wb}}$ is the RMS voltage applied on the Wheatstone bridge, *G* is the gain of the full acquisition system, $s_{\mathrm{wb}}$ is the sensitivity of the Wheatstone bridge, and $T_{0}$ is the center temperature offset of the current scale. The divider of this expression is nonlinear in reality, and needs to be calibrated for the whole sensing range. Most importantly, as shown in the following sections, this approximation allows the expected theoretical noise of our measurement system and temperature coefficient of the read-out to be assessed.

#### 2.3.2. Noise Analysis

The full chain noise was theoretically analysed using the values of the datasheets of the different components and compared with experimental results. All the noise values used in this section always refer to the input of the chain, i.e., in reference to the sensor itself, and are converted to temperatures with a full expression similar to Equation (Equation 2). In addition, when adding different noise sources, we consider all of them uncorrelated; hence, we use the square root of the sum of squares.

The first element to consider is the noise of the sensor resistor divider. In this case, in principle we expect only thermal noise, which is provided by the following expression:(3)$$S_{\mathrm{sense}}^{1/2}\left(f\right)=\sqrt{4K_{b}TR_{\mathrm{eq}}}$$
where $K_{b}$ is the Boltzmann constant, *T* is the temperature in Kelvin, and $R_{\mathrm{eq}}$ is the equivalent parallel resistance between the thermistor and the temperature-stable resistor. At the center of the scale, at 25 °C, for the 30 kΩ fixed resistor and thermistor we have a noise of 16 $\mathrm{nV}/\sqrt{\mathrm{Hz}}$; using the previous sensitivity, this translates to a noise density of 0.5 µ$K/\sqrt{\mathrm{Hz}}$. In contrast, because the reference arm uses two resistors of one kilo-ohm each, the equivalent thermal noise is only 0.1 µ$K/\sqrt{\mathrm{Hz}}$.

It is important to note that all the fixed resistances were obtained using metal foil compositions with 0.05 ppm/K thermal coefficients in order to reduce the temperature sensitivity of the read-out electronics as much as possible (described in detail in the next section) and low tolerance to reduce the overall system random uncertainty [25].

For voltage difference amplification, the following equation describes its noise [17]:(4)$$S_{\mathrm{OA}}^{1/2}\left(f\right)=\sqrt{i_{\mathrm{OA}}^{2}(R_{\mathrm{ref}}^{2}+R_{\mathrm{sense}}^{2}\left(T\right)+2e_{\mathrm{OA}}^{2}+8k_{B}TR_{f}/G}$$
where $i_{\mathrm{OA}}$ and $e_{\mathrm{OA}}$ are the current noise density and voltage noise density, respectively, of the operational amplifiers, $R_{\mathrm{ref}}$ and $R_{\mathrm{sense}}$ are the respective equivalent resistances of the reference arm and the sensing arm of the Wheatstone bridge, $R_{f}$ is the feedback resistance of the amplifiers, and *G* is the gain during this stage. For our design, this results in 0.3 µ$K/\sqrt{\mathrm{Hz}}$ of noise.

Now, we have the fully differentiated amplifier noise. In this case, it is provided by the following expression:(5)$$S_{\mathrm{FDA}}^{1/2}\left(f\right)=\frac{1}{G}\sqrt{2{\left(i_{\mathrm{FDA}}R_{\mathrm{FDA}}\right)}^{2}+{\left(2e_{\mathrm{FDA}}\right)}^{2}+16k_{B}TR_{f,\mathrm{FDA}}}$$
where $i_{\mathrm{FDA}}$ and $e_{\mathrm{FDA}}$ are the current noise density and voltage noise density of the FDA, respectively, $R_{\mathrm{FDA}}$ is the equivalent input resistance seen by the FDA, and $R_{f,\mathrm{FDA}}$ is the feedback resistance of the FDA. In this case, it results in a contribution of 0.15 µ$K/\sqrt{\mathrm{Hz}}$ at the 10 Hz modulation frequency.

Finally, we have the ADC, which has two contributions, the quantization noise and the conversion noise. The quantization noise is provided by
(6)$$S_{\mathrm{ADC},q}^{1/2}\left(f\right)=\frac{1}{G}\frac{V_{\mathrm{FS}}/2^{n}}{\sqrt{12}}\frac{1}{\sqrt{f_{s}/2}}$$
where $V_{\mathrm{FS}}$ is the full scale input voltage of the ADC, *n* is the number of bits, and $f_{s}$ is the sampling frequency used (in the tenth of kHz range in our case). In this case, the quantization noise is mostly negligible due to the high-resolution and sampling frequency, contributing only 0.02 µ$K/\sqrt{\mathrm{Hz}}$ of noise.

For the conversion noise, we use the following expression:(7)$$S_{\mathrm{ADC},c}^{1/2}\left(f\right)=\frac{1}{G}\frac{\sigma }{\sqrt{f_{s}/2}}$$
where σ is the RMS noise provided by the datasheet of the device. In our case, we have an ADC conversion noise of 0.29 µ$K/\sqrt{\mathrm{Hz}}$

If we add all the contributions as uncorrelated noise, we have a total noise projection at the center of the scale of 0.8 µ$K/\sqrt{\mathrm{Hz}}$, below our noise aim of 1 µ$K/\sqrt{\mathrm{Hz}}$. However, because the sensitivity degrades when moving too far from the center, for the highest temperature it reaches almost 0.9 µ$K/\sqrt{\mathrm{Hz}}$. The result for the full temperature range can be seen in Figure 5, and a summary of the contributions is shown in Table 1. It can be seen from the summary table that the dominant contribution element is the sensor arm itself, which is the ideal case. Nonetheless, the value is not very high, meaning that there is room for improvement if the only focus were the noise floor. For instance, if the noise floor were reduced further, the gain of the chain could be easily increased and the ADC noise contribution at least reduced. This obviously would decrease the sensing range, which is another requirement with which we need to comply. In summary, the presented design is, as always, a trade-off between different objectives, and we can see from the theoretical noise that even with this relatively large temperature range, we are theoretically compliant with the noise objective.

### 2.4. Temperature Coefficient

Due to the high-precision aim of this project, thermal noise fluctuations on the read-out itself may couple to the measurements. Hence, any thermal fluctuations at the acquisition chain produce a change in the temperature reading as well, and this is indistinguishable from any change in the temperature at the sensor placement.

Taking into account the temperature coefficients of each element of the chain, the voltage sampled at the ADC can be written as
(8)$$V_{\mathrm{ADC}}\left(t\right)=V_{\mathrm{wb}}\left(t\right)G\left(1+\alpha_{G}\Delta T_{a}\right)\left[\alpha_{\mathrm{wb}}\Delta T_{a}+s_{\mathrm{wb}}\left(T_{s}-T_{0}\right)\right]$$
where $\alpha_{G}$ is the thermal gain error coefficient, $T_{a}$ is the ambient temperature at the electronics, $\alpha_{\mathrm{wb}}$ is the zero error thermal coefficient, and $T_{s}$ is the temperature at the sensor. Because we use high-stability resistances, the zero thermal error coefficient in our design is very low, equal to 367 nV/K. For the gain error $\alpha_{G}$, the dominating component is the ADC gain drift; hence, we can assume $\alpha_{G}≃\alpha_{\mathrm{ADC}}=1.3\;\mathrm{ppm}/K$.

Furthermore, as the gain error depends on the temperature point being sensed, it becomes important when sensing large temperature ranges near the edges of the range.

Introducing the previous equation in Equation (Equation 2) and transforming the expression into spectral densities, we have
(9)$$S_{T_{m},T_{a}}\left(f\right)=\left[\frac{\alpha_{\mathrm{wb}}^{2}}{s_{\mathrm{wb}}^{2}}+\alpha_{\mathrm{ADC}}^{2}{\left(T_{s}-T_{0}\right)}^{2}\right]S_{T_{a}}\left(f\right)$$
where $S_{T_{a}}\left(f\right)$ are the read-out ambient temperature fluctuations and $S_{T_{m},T_{a}}\left(f\right)$ is the measured noise from these ambient temperature fluctuations.

Finally, we can combine the ambient temperature effects and read-out noise as follows:(10)$$S_{T_{m}}\left(f\right)=S_{\mathrm{read}-\mathrm{out}}\left(f+f_{0}\right)+\left[\frac{\alpha_{\mathrm{wb}}^{2}}{s_{\mathrm{wb}}^{2}}+\alpha_{\mathrm{ADC}}^{2}{\left(T_{s}-T_{0}\right)}^{2}\right]S_{T_{a}}\left(f\right)$$
where $S_{\mathrm{read}-\mathrm{out}}\left(f\right)$ is the noise from the read-out itself (found in the previous section) and $f_{0}$ is the modulation frequency of the modulation signal. It is important to note that while the read-out noise is frequency-shifted thanks to the modulation, in the case of the temperature fluctuations coupled to the gain (specifically, due to the gain being multiplying and not adding to our measurement), we need to consider the effect at low frequencies down to 0.1 mHz. The worst-case situation is at the edge of the scale, where the sensed temperature may differ from the center of the scale by up to 15 K. Furthermore, the electronics noise at this point reaches almost 0.9 µ$K/\sqrt{\mathrm{Hz}}$. Taking all this into account, the maximum thermal fluctuations on the read-out need to be below 22 $\mathrm{mK}/\sqrt{\mathrm{Hz}}$.

This requirement can be demanding assuming ambient temperature fluctuations of a typical laboratory environment, as shown in the results (Section 3). However, it is worth noting that this limiting factor imposes restrictions mainly during on-ground testing; conditions in space are typically milder in this sense. For instance, for LISA Pathfinder, the environmental thermal fluctuations at the electronics locations are below 2 $\mathrm{mK}/\sqrt{\mathrm{Hz}}$ at 1 mHz [18], well below this requirement.

## 3. Results and Discussion

The test bench was designed to achieve a temperature stability of $100\;\mathrm{nK}/\sqrt{\mathrm{Hz}}$ in its interior down to 1 mHz, allowing for a relaxation factor at lower frequencies. As this design goal is below the performance of our sensors, the validation strategy cannot rely on direct measurements. Instead, the test bench can be characterised by its capability to suppress external environment temperature fluctuations, which is fully determined by its thermal transfer function.

In order to obtain a precise estimate of this transfer function, we took advantage of the Peltiers attached to the external thermal shield. We injected a sum of sinusoids to these actuators that induced a temperature modulation in the external layer of our thermal shields. This modulation was then measured, while highly suppressed, in the inner block of our setup. The transfer function between sensors in the outer shield and the inner aluminium core is shown in Figure 6. We identified the frequencies in the plot where the injections were applied. Despite the thermal injections being strongly attenuated inside through the thermal shields, the highly sensitive temperature read-out in this frequency regime enables precise estimation of the thermal transfer function. The analytical transfer function of the test bench has a complicated mathematical expression, typically represented by an infinite sum of low passes [19,22]. We can, however, obtain very good estimates for our purposes by approximating this behaviour to a second-order transfer function, shown Figure 6 and labelled as “Theoretical”. The differences between the infinite sum and second-order approaches are, however, appreciable only in the high-frequency tail of the low pass.

The most demanding requirement for our experiment is the achievement of the noise floor of 1 µ$K/\sqrt{\mathrm{Hz}}$ down to 1 mHz, with a relaxation factor at low frequencies. In order to evaluate this requirement, we undertook a series of long-term runs consisting of several days of temperature acquisition, while the test bench active temperature control was used to stabilise the inner core of the experiment.

Figure 7 shows the typical performance obtained with a representative subset of the sensors. The noise floor of the read-out system reaches 1 µ$K/\sqrt{\mathrm{Hz}}$ in most of the band, and even below this for frequencies above 10 mHz. The requirement is just as well fulfilled in the lower frequency regime below 1 mHz, a highly demanding regime for the electronics read-out.

The vicinity of the millihertz region, however, is where we find a slight excess noise above our requirements, typically reaching 2–3 µ$K/\sqrt{\mathrm{Hz}}$. We have undertaken an extensive noise hunting campaign to track down the potential contributions limiting the performance in this frequency regime. Among others, we evaluated the cross-spectrum between the different sensors in the aluminium core, which helped us determine that the excess noise in this frequency band has a common origin between the different sensors. Moreover, the analysis showed that this excess noise was external to our read-out.

The first potential external contribution to discard was that originating from environmental temperature fluctuations entering into our test bench. To discard this possibility, we took advantage of the thermal transfer function evaluated in the previous section to estimate the noise projection of these environment fluctuations after they reach the inner core of the test bench. Figure 7 shows that the expected noise projection does not disturb the performance of our sensor. Hence, we can conclude that the test bench performs according to design, and no excess temperature noise is entering the setup.

Considering that the excess noise appears to be correlated with all the sensors, we looked for a second source of coupling within the environment. A natural candidate for this was the temperature fluctuations in the environment coupling through the thermal coefficient of our read-out electronics, as seen in Section 2.4. To evaluate this contribution experimentally, we injected controlled heat inputs into our read-out while the electronics measured the high-stability resistance instead of the temperature. The measured response was therefore from variations induced in the electronics, which translated into an apparent temperature change. In this way, we were able to experimentally determine the temperature coupling coefficient of the read-out to environment temperature fluctuations. The obtained coupling coefficient is up to 52 ppm/K for the unbalanced Wheatstone bridge configuration. This value is twice that of the previously determined theoretical value of the temperature coefficient for the read-out. Our best guess is that the ADC gain drift is in reality much worse than the nominal value reported in the datasheet used for the theoretical calculations. With these values in hand, we were able to estimate the noise contribution arising due to this effect. As shown in Figure 7, labelled as “FEE temperature coupling”, the expected contribution from this noise coupling dominates our measurements in the frequency regime where we found the excess noise.

## 4. Conclusions

In this paper, we have presented a design for a temperature measurement subsytem able to reach 1 $\mu K/\sqrt{\mathrm{Hz}}$ in most of the measuring band down to 1 mHz, and to reach 20 $\mu K/\sqrt{\mathrm{Hz}}$ at 0.1 mHz. The presented scheme is based on resistive sensors in a Wheatstone bridge configuration, which is AC modulated to minimize the 1/f noise. The design is compact and the components are compatible with its implementation in a space mission.

In addition, we have described a test bench designed to achieve a sufficiently stable environment for our temperature subsystem to be put to the test. This facility is composed of a vacuum tank with a series of passive thermal shields acting as thermal low-pass filters, along with Peltier elements used in an active closed loop and for calibration purposes. During our tests, we confirmed that the test bench can guarantee the required temperature stability needed to characterise our temperature measurement subsystem. In fact, we were able to indirectly estimate that the stability achieved in the inner core of the test bench is below 100 $\mathrm{nK}/\sqrt{\mathrm{Hz}}$ at 1 mHz.

During the testing phase, we analysed the origin of excess noise in the millihertz frequency range. Our analysis points towards a coupling of the read-out electronics with the environment temperature fluctuations through the read-out electronics temperature coefficient. This is a limiting factor that could be attenuated by either improving the electronics thermal isolation (actively or passively) or by imposing stringer requirements on the lab environment. We note, however, that while this noise contribution is a limiting factor that needs to be addressed during on-ground testing, it does not restrict the performance of the temperature measurement subsystem during in-flight operations.

The main motivation behind this development is the design of a prototype temperature diagnostics subsystem for the forthcoming LISA space-borne gravitational wave detector. Nevertheless, the design is generic enough to find application in many other areas with demanding requirements in terms of high-precision temperature measurements.

## Figures and Tables

**Figure 1 sensors-23-00145-f001:**
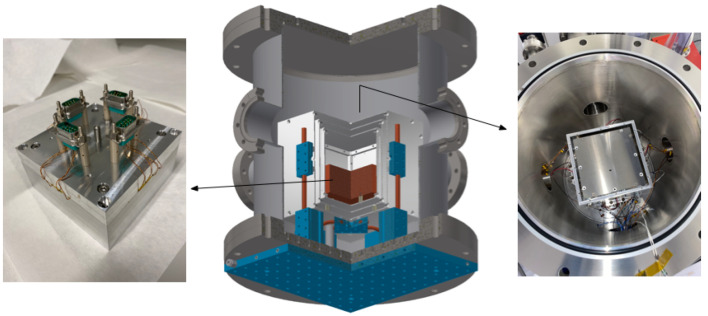
(**Center**): Vacuum chamber (dark gray) and thermal shields (light gray) drawing. The aluminium block where the sensors are placed is the central element, colored in brown. In blue, the water-cooled breadboard and heat pies used to extract the heat of the Peltier elements can be seen. (**Left**): Picture of the aluminium block hosting the sensors. (**Right**): Top view of the inside of the vacuum chamber.

**Figure 2 sensors-23-00145-f002:**
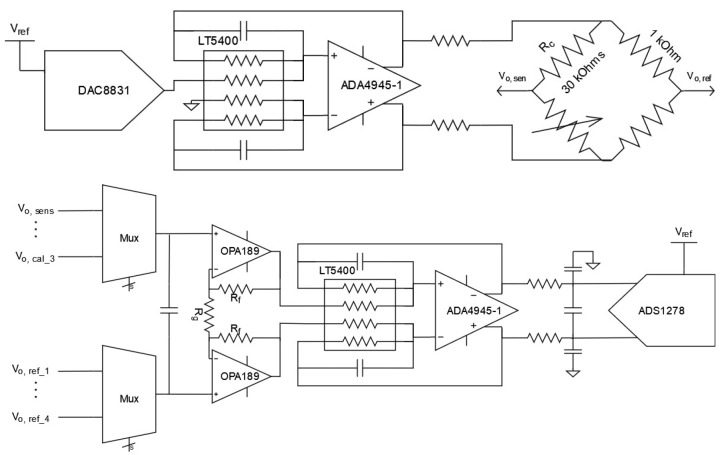
(**Top**): Modulation circuit with Wheatstone bridge. (**Bottom**): Acquisition chain of the output of the Wheatstone bridge.

**Figure 3 sensors-23-00145-f003:**
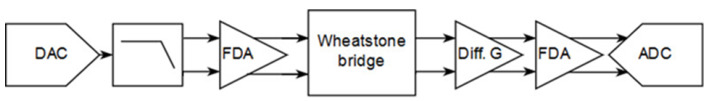
Read-out block diagram.

**Figure 4 sensors-23-00145-f004:**
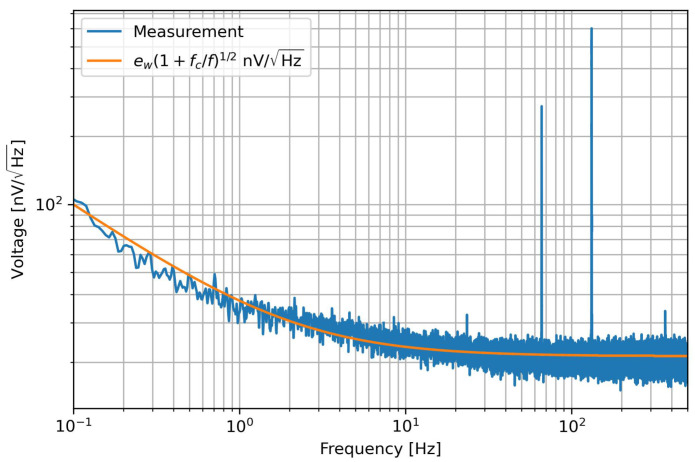
Noise of the electronics without modulation. The values of the fit are $e_{w}=$ 21 $\mathrm{nV}/\sqrt{\mathrm{Hz}}$ and $f_{c}=$ 2 Hz. The peaks seen at around 100 Hz are interference.

**Figure 5 sensors-23-00145-f005:**
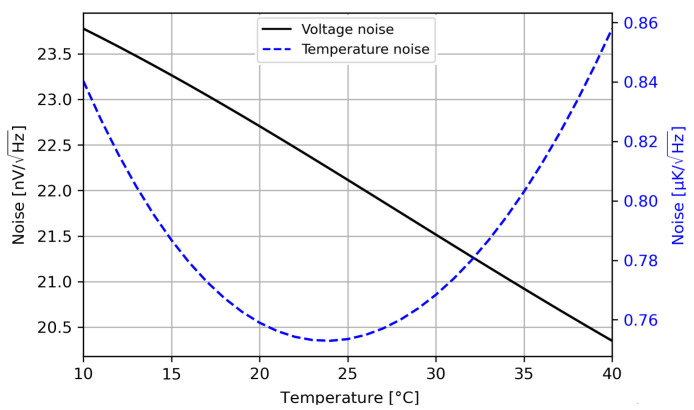
Theoretical noise in the full sensing range; the black solid line is the voltage noise, and the dashed blue line is the same noise converted to temperature by means of the expected sensitivity.

**Figure 6 sensors-23-00145-f006:**
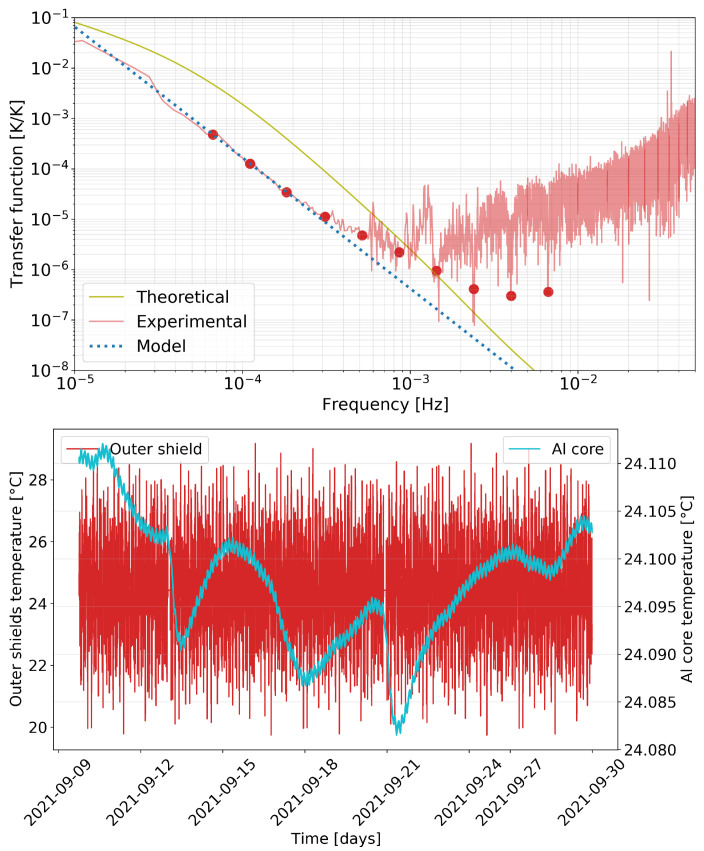
Evaluation of the thermal transfer function of the test bench. (**Top**): Experimental estimation of the thermal transfer function; the dots mark the frequencies where temperature injections were applied and the dotted line shows the fit (labeled as Model) to these frequencies. (**Bottom**): Detail of the time series during the evaluation of the test bench thermal transfer function; the temperature measured in one sensor of the Al core appears strongly suppressed when compared to the measurement in the outer shield.

**Figure 7 sensors-23-00145-f007:**
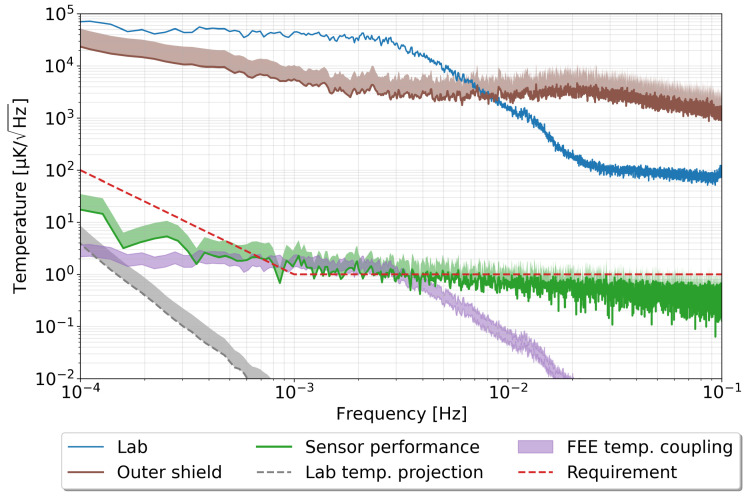
Performance of the temperature measurement system. A sensor attached to the external wall of the vacumm chamber describes the temperature fluctuations in the lab (`Lab’). The brown curve (`Outer shield’) shows the mean of the ten sensors attached to the external layer of the three layers composing the thermal shield inside the vacuum chamber. The green curve (`Sensor performance’) is the mean of three NTC sensors inside attached to the aluminium core, and the grey curve (`Lab temperature projection’) shows the noise projection of the outer shield fluctuations inside the aluminium core. The purple area (`FEE temperature coupling’) shows the estimate corresponding to the coupling of lab temperature fluctuations through the read-out electronics temperature coefficient. Whenever several sensors were used to build the curve, we show the standard deviation of the measurements as a shaded area marking the upper limit.

**Table 1 sensors-23-00145-t001:** Summary of the read-out noise contributions.

	Temperature Noise Density	Contribution on the Overall
	**[** $\mu K/\sqrt{\mathrm{Hz}}$ **]**	**[%]**
Sensor Arm	0.5	57
Reference Arm	0.1	2
Difference Amplification	0.3	18
Fully Differential Amplifier	0.2	4
Analog-to-digital converter	0.3	19

## Data Availability

The data presented in this study are available on reasonable request from the corresponding author.

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
