# Peer review of "Resistive-Based Micro-Kelvin Temperature Resolution for Ultra-Stable Space Experiments"

_sensors, 2022, doi:10.3390/s23010145_

Round 1
Reviewer 1 Report
The authors are describing a resistive sensor placed within a Wheatstone bridge configuration for high precision temperature measurements for space applications primarily for gravitational wave detectors. The paper is of good quality and is well-written and as such could be considered for publication in MDPI Sensors.
I would have the following remarks.
Besides 3-D Model shown in Figure 1, please include a representative photo of your test setup including the photos of the sensor itself.
The authors claim that all COTS components are used in the design. Please also specify exactly which other components (besides the TI's ADS1278 ADC) are used, in particular DAC, the (operational) amplifiers, the low-pass filter, etc. Please also draw a more detailed circuit schematic, since the block-level one on Fig. 2 is insufficient. The schematic should include the Wheatstone bridge with resistance values as they are given inside section 2.3.2 on noise analysis.
Is the 10Hz up-conversion modulation frequency sufficient to avoid the pink noise? Is it indeed the optimal one? I wonder if the results would look better for 100Hz or 1kHz frequencies for example? I know it would require a lot of time to perform these measurements, but having it done across multiple modulation frequencies, and without modulation altogether would give an important insights where the measurement sweet spot lies.
Even though square signal modulation has it merits over the sinusoidal one, it also has some drawbacks too. Please elaborate.
Please put a legend in Figure 3 which labels dashed and solid lines and put this explanation in the figure caption, too.
Good paper overall.
Author Response
The authors are describing a resistive sensor placed within a Wheatstone bridge configuration for high precision temperature measurements for space applications primarily for gravitational wave detectors. The paper is of good quality and is well-written and as such could be considered for publication in MDPI Sensors.
Thank you very much for your kind words. Thanks also for your comments which we think have helped us to improve the paper quality.
I would have the following remarks.
Besides 3-D Model shown in Figure 1, please include a representative photo of your test setup including the photos of the sensor itself.
We added two photos in Figure 1, one of the whole chamber and another of the aluminium cube housing the sensors.
The authors claim that all COTS components are used in the design. Please also specify exactly which other components (besides the TI's ADS1278 ADC) are used, in particular DAC, the (operational) amplifiers, the low-pass filter, etc. Please also draw a more detailed circuit schematic, since the block-level one on Fig. 2 is insufficient. The schematic should include the Wheatstone bridge with resistance values as they are given inside section 2.3.2 on noise analysis.
Added in the text of section 2.3 the specific components used and added Figure 3 with a schematic depiction of the circuit, also including the component's models.
Is the 10Hz up-conversion modulation frequency sufficient to avoid the pink noise? Is it indeed the optimal one? I wonder if the results would look better for 100Hz or 1kHz frequencies for example? I know it would require a lot of time to perform these measurements, but having it done across multiple modulation frequencies, and without modulation altogether would give an important insights where the measurement sweet spot lies.
To verify which is the minimum frequency to go outside of the 1/f dominated region, as the reviewer suggest, you need to measure the noise floor of the electronics without any modulation. The result can be seen in the newly added Figure 4. As you can see, there is no significant change for our electronics if you go much above 10 Hz. An explanation about it has been also included at the beginning of section 2.3.1 We also agree that for a full characterization the temperature measurement should be done also at different modulation frequencies. But a part of the time it requires, as the reviewer also mentions, is the fact that it should be done with a representative harness (length, insulation material, connector, etc) that currently is not known for the LISA mission. Hence we kept it at this first level which we think already point that the best frequency is most probably around the 10 Hz value.
Even though square signal modulation has it merits over the sinusoidal one, it also has some drawbacks too. Please elaborate.
Added discussion about the advantages of the sinusoidal over the square wave in section 2.3.1, lines 193 to 210.
Please put a legend in Figure 3 which labels dashed and solid lines and put this explanation in the figure caption, too.
Fixed legend and caption for Figure 3 (now Figure 5).
Good paper overall.
Again, thanks for you kind works and review. And we hope that we correctly addressed all the points raised by you.
Reviewer 2 Report
I enjoyed reading this work. Although the manuscript is theoretical, it offers some new insight about the model proposed. I have simple comments:
1-The stability conditions are very important. I think more discussions are required. Besides, the stability against fluctuations and perturbations is very important to mention. This require analytical studies. They were done in this manuscript, obviously, but it is important to discuss it.
2-Some missing references:
Meas. Sci. Technol. 2019, 30, 112001; Nano Lett. 2018, 18, 7165; ACS Omega 2021, 6, 23052−23058; Journal of Thermal Stresses 2022 45 (4), 303-318
Author Response
I enjoyed reading this work. Although the manuscript is theoretical, it offers some new insight about the model proposed. I have simple comments:
We appreciate the referee’s comment. We would like to stress however that the work in the manuscript is experimentally based. The theoretical analysis in the paper are fundamentally oriented to describe the design that we latter experimentally test.
1-The stability conditions are very important. I think more discussions are required. Besides, the stability against fluctuations and perturbations is very important to mention. This require analytical studies. They were done in this manuscript, obviously, but it is important to discuss it.
In Sec. 2.1 we provided an analysis of how the experiment is thermally isolated with respect the environment fluctuations. We provide as well references of previous work with regards long term stability experiments. The analysis is further supported with an experimental verification of the thermal transfer function of our test bench in Fig. 8.
In what refers the stability and dependence to environmental fluctuations of our read-out electronics, we derive in sec 2.4 the thermal coefficient of this read-out. As we discussed in our conclusions, this analysis was also experimentally tested. The contribution of environmental fluctuations to our measurement is characterised in Fig. 9 where we show that, in a certain regime, this is in fact limiting the performance of our system.
2-Some missing references:
Meas. Sci. Technol. 2019, 30, 112001; Nano Lett. 2018, 18, 7165; ACS Omega 2021, 6, 23052−23058; Journal of Thermal Stresses 2022 45 (4), 303-318
We appreciate the suggestion from the referee. We have included those references that we could motivate in the framework of the paper. For others, unfortunately, we did not find a natural link with the work described in our manuscript.